# Prognostic Factors for Localized Clear Cell Renal Cell Carcinoma and Their Application in Adjuvant Therapy

**DOI:** 10.3390/cancers14010239

**Published:** 2022-01-04

**Authors:** Kalle E. Mattila, Paula Vainio, Panu M. Jaakkola

**Affiliations:** 1Department of Oncology and Radiotherapy, FICAN West Cancer Centre, University of Turku, Turku University Hospital, Hämeentie 11, 20521 Turku, Finland; panu.jaakkola@tyks.fi; 2Department of Pathology, University of Turku, Turku University Hospital, Hämeentie 11, 20521 Turku, Finland; paula.vainio@tyks.fi

**Keywords:** adjuvant therapy, clear cell renal cell carcinoma, biomarker, prediction model, prognostic factor

## Abstract

**Simple Summary:**

Approximately one fifth of patients with newly diagnosed renal cell carcinoma (RCC) present with metastatic disease and over one third of the remaining patients with localized RCC will eventually have metastases spread to distant sites after complete resection of the primary tumor in the kidney. Usually, disease recurrence is observed within the first five years of follow-up, but late recurrences after five years are seen in up to 10% of patients. Despite novel biomarkers, simple histopathological factors, such as tumor size, tumor grade, and tumor extension into the blood vessels or beyond the kidney, are still valid features in predicting the risk of disease recurrence after surgery. The optimal set of prognostic factors remains unclear. The results from ongoing placebo-controlled adjuvant therapy trials may elucidate prognostic features that help to define high-risk patients for disease recurrence.

**Abstract:**

Approximately 20% of patients with renal cell carcinoma (RCC) present with primarily metastatic disease and over 30% of patients with localized RCC will develop distant metastases later, after complete resection of the primary tumor. Accurate postoperative prognostic models are essential for designing personalized surveillance programs, as well as for designing adjuvant therapy and trials. Several clinical and histopathological prognostic factors have been identified and adopted into prognostic algorithms to assess the individual risk for disease recurrence after radical or partial nephrectomy. However, the prediction accuracy of current prognostic models has been studied in retrospective patient cohorts and the optimal set of prognostic features remains unclear. In addition to traditional histopathological prognostic factors, novel biomarkers, such as gene expression profiles and circulating tumor DNA, are extensively studied to supplement existing prognostic algorithms to improve their prediction accuracy. Here, we aim to give an overview of existing prognostic features and prediction models for localized postoperative clear cell RCC and discuss their role in the adjuvant therapy trials. The results of ongoing placebo-controlled adjuvant therapy trials may elucidate prognostic factors and biomarkers that help to define patients at high risk for disease recurrence.

## 1. Introduction

Renal cell carcinoma (RCC) is the third most common newly diagnosed urogenital cancer after prostate and bladder cancer. In 2020, the number of new kidney cancer diagnoses was over 400,000 and it caused nearly 180,000 deaths worldwide [1]. The most prevalent histological subtype, clear cell renal cell carcinoma (ccRCC), accounts for 75–80% of all RCCs and has been associated with inferior survival compared to papillary (10–15%) and chromophobe (5%) RCCs [2]. Localized RCC can be treated with curative intent by radical (RN) or partial nephrectomy (PN). PN is preferred for smaller tumors (T1–2N0M0) if technically feasible without compromising the oncological outcome of surgery (negative surgical margins). Small renal tumors might be eligible for radiofrequency ablation. Lymph node dissection (LND) is not routinely performed unless there is a suspicion of metastatic lymph nodes preoperatively or during surgery. If macrovascular invasion is present, tumor thrombus is removed from the renal and caval vein during surgery [3,4].

Unfortunately, approximately 20% of RCC patients present with primarily metastatic disease and over one third of patients will eventually develop distant metastases [5]. Despite recent advances in the medical treatment of advanced RCC (antiangiogenic receptor tyrosine kinase inhibitors (TKI), immune checkpoint inhibitors (ICI) and TKI–ICI combinations), metastatic disease will, in most cases, lead to death. Individualized, risk-based, regular imaging follow-up after surgery for localized RCC for at least five years with thoracic and abdominal CT is recommended to detect disease progression (local recurrence or distant metastases) early. If detected early with few or solitary metastases, the patient may still be curable surgically or with high-dose radiation therapy and may be eligible for oncologic therapies [3,4]. The aim of this review is to provide an overview of the clinical prognostic models for localized ccRCC. Understanding the risk of disease progression after surgery of localized disease is essential for designing postoperative follow-up as well as for designing adjuvant drug trials in localized ccRCC.

## 2. Histopathological and Clinical Prognostic Factors for Localized ccRCC

The TNM classification of malignant tumors (American Joint Committee on Cancer (AJCC)) has been used since 1977 as a prognostic staging system for multiple solid tumors [6]. The staging of renal cell carcinoma based on pathologic examination and radiological imaging provides crucial prognostic information. Stage I (T1N0M0: tumor ≤ 7 cm) and stage II (T2N0M0: tumor > 7 cm) tumors are limited to the kidney, whereas stage III (T3N0, T1−3N1: tumor invades renal vein, perinephric tissues, or presents with regional lymph node metastases) and stage IV (T4N_any_M0, T_any_N_any_M1: tumor extends beyond Gerota fascia or presents with distant metastases) tumors extend beyond the kidney [7]. In 1993–2004, 54.7%, 10.6%, 16.1%, and 18.6% of ccRCC tumors in the National Cancer Database were classified as stage I, II, III, and IV, respectively [8]. Stage I and II RCCs had significantly better 5-year survival rates (90.4% and 83.4%) compared to stage III and stage IV RCCs (66.0% and 9.1%) [8]. In a 2004–2015 Surveillance, Epidemiology, and End Results (SEER) database cohort (77% had ccRCC), the pathologic TNM stage was I (64.3%), II (10.9%), III (16.8%), and IV (8%) and the 5-year survival rates after nephrectomy were 97.4%, 89.9%, 77.9%, and 26.7% for stage I, II, III, and IV RCCs, respectively [9]. The proportion of stage I tumors has increased, probably due to the incidental detection of small renal tumors in abdominal imaging studies [8]. The increase in the survival rate of stage III and IV tumors is probably driven by VEGF-targeted TKI therapies introduced in the treatment of advanced RCC in the 2000s.

### 2.1. Microscopical Histopathological Prognostic Factors

In addition to the TNM stage, several histopathological factors affect the prognosis of localized ccRCC patients. Numerous tumor grading systems have been introduced to assess the histological differentiation of RCC cells. The Fuhrman and the WHO/ISUP grading systems are the most widely used. In 1982, Fuhrman developed a four-tiered grading system that is based on the assessment of nuclear size, nuclear shape, and nucleolar prominence. The estimated 5-year survival rate of RCC patients was 64% (grade I), 34% (grade II), 31% (grade III), and 10% (grade IV) [10]. In 2012, the International Society of Urological Pathology reformed the four-tiered grading system based on the prominence of nucleoli (grades 1–3) and grade 4 tumors showing extreme tumor nuclear pleomorphism, giant cells, or sarcomatoid/rhabdoid dedifferentiation [11].

Approximately 5% of RCCs undergo epithelial to mesenchymal transition and present with sarcomatoid differentiation, and sarcomatoid features have been observed in clear cell, papillary, and chromophobe RCCs [12]. Sarcomatoid morphology is associated with more aggressive cancer behavior: sarcomatoid RCCs (sRCCs) often present with a bulky primary tumor (higher size and stage) and higher tumor grade [12,13,14]. Metastases are seen in as much as 60–80% of newly diagnosed cases [12] and close to 80% of patients with localized sarcomatoid RCC have been observed to develop disease recurrence within two years after nephrecotomy [15,16]. Sarcomatoid RCCs have unfavorable prognosis compared to ccRCCs regardless of tumor stage: 5-year cancer-specific mortality estimates were 32%, 63%, and 82% for stage I–II, III, and IV sRCCs, compared to 6%, 20%, and 64% for stage I–II, III, and IV ccRCC patients [14].

Tumor necrosis is another established adverse histological feature in RCC. It is associated with a larger tumor size, higher grade, and higher proliferative activity, and it is considered to indicate biologically aggressive tumor behavior [17,18]. The presence of tumor necrosis has been reported in 21–32% of ccRCCs [19,20] and it has also been associated with inferior survival outcomes in multiple studies [20,21,22,23,24,25]. The combination of WHO/ISUP grading and tumor necrosis outperformed WHO/ISUP grading after adjusting for TNM stage. Researchers observed that the presence of tumor necrosis affected the prognosis, especially in WHO/ISUP grade 3 tumors. The 10-year cancer-specific survival was 62% in grade 3 tumors without necrosis but only 30% in grade 3 tumors with necrosis [26].

RCCs are highly vascularized, and microscopic vascular invasion is observed in 5.6–45% of tumors [19]. Tumor cells can spread via blood and lymph vessels to distant sites (lungs, bones, liver, etc.) and lymph nodes. Microvascular invasion is defined as tumor cells within small vessels in the tumor pseudocapsule, tumor, or renal parenchyma adjacent to the tumor [27]. Microvascular invasion (MVI) was more commonly present in ccRCC (29%) than in non-ccRCC (12%) and it was associated with metastatic spread and inferior survival in ccRCC patients [28]. MVI was found to be associated with a larger tumor size, higher Fuhrman grade, more advanced T stage, the presence of lymph node and distant metastases, as well as a shorter survival time in univariate but not in multivariate analysis [29]. In another cohort of RCC patients (93% had ccRCC), MVI was observed to correlate with metastases and shorter disease-free survival as well as cancer-specific survival [30].

Partial nephrectomy (PN) is the standard of treatment for small renal tumors if technically feasible. Positive surgical margins (PSM) have been observed in up to 18% of patients after surgery for localized RCC [31,32,33,34]. However, the effect of PSM on oncologic outcomes (recurrence-free and cancer-specific survival) is controversial. Local tumor recurrence in the surgical bed is uncommon. In a retrospective study, local tumor bed recurrence was found in only 1.9% of patients who underwent PN, and PSM were found in 15.9% of patients with local tumor bed recurrence, compared to 3% in the control group [35]. However, PSM is more common in patients with other adverse features (higher tumor stage, grade, multiple tumors, solitary kidney) and local recurrences are also observed in patients with negative surgical margins [32,33,35]. Therefore, imaging surveillance is preferred over radical nephrectomy in patients with PSM after PN.

### 2.2. Macroscopical Histopathological Prognostic Factors

Tumor extension into perirenal tissues, renal vein, and regional lymph nodes might be discovered in preoperative radiological imaging or during surgery but sometimes only after microscopical evaluation of resected tumor and regional lymph nodes. Tumor invasion into perirenal tissues (perirenal fat, renal sinus fat) or macrovascular invasion into the renal vein and inferior vena cava (IVC) or local lymph nodes (T3N0, T1–3N1, stage III) lead to inferior oncologic outcomes compared to stage I and II tumors [8,9]. Perinephric fat, renal sinus fat, and renal vein invasion were present in 26%, 9%, and 29% of T3a tumors and patients with multiple extrarenal extensions had inferior progression-free and overall survival [36]. The association of concomitant fat invasion and renal vein invasion with poorer cancer-specific survival has also been observed in other studies [37,38,39]. Upper pole RCCs may invade directly into the adrenal gland. These tumors are classified as T4 as well as tumors extending beyond Gerota’s fascia, leading to worse oncologic outcomes compared to T1–3N0/Nx tumors [8,9].

Tumor extension into the renal vein and inferior vena cava has been observed in 23% and 7–13% of patients, respectively [40,41]. Patients with venous invasion had significantly shorter survival compared to tumors limited to the kidney [41]. The Mayo Clinic’s thrombus classification is commonly used to classify the level of tumor extension into the IVC [42]. The prognostic significance of the tumor thrombus level is controversial. In a retrospective multicenter evaluation of tumor thrombus level, 78%, 16%, and 5% of the patients had tumor extension into the renal vein, IVC below diaphragm, and above diaphragm, respectively [43]. The level of tumor thrombus in the IVC (below or above diaphragm) was not statistically significantly associated with the survival time, but patients with tumor thrombus in the IVC had shorter survival (18–26 months) compared to patients with tumor thrombus in the renal vein (52 months) [41,43]. In another multicenter study (89.9% had ccRCC), a higher tumor thrombus level was independently associated with shorter cancer-specific survival [44]. Notably, the patients in these studies were treated before modern TKI and ICI therapies. In a contemporary analysis of 6340 patients who underwent surgery for localized RCC (93.4% had ccRCC), only 3.6% of the patients had venous tumor thrombus and the level of thrombus was not associated with the risk of recurrence or death [45].

Lymph node dissection (LND) is not routinely performed during nephrectomy unless there is a suspicion of metastatic lymph nodes preoperatively or during surgery, and local lymph node status usually remains unknown (Nx). LND has not proven therapeutic but is a prognostic procedure to assess metastatic spread to regional (hilar, abdominal, para-aortic, and para caval) lymph nodes. In the SEER database analysis, 24.8% of patients (59.4% had ccRCC) underwent lymph node dissection (LND) and metastatic lymph nodes were observed in 17.1% (9.3% of T2 and 21.6% of T3) of the patients who underwent LND [46]. In another study, local lymph node metastases were present in 11% of nonmetastatic RCC patients (90.7% had ccRCC) who underwent nephrectomy [47]. The patients with regional lymph node metastases (T1–3N1) have as poor survival as stage IV patients [47,48]. The 5-year survival rates were 61.9%, 22.7%, and 15.6% for stage III lymph node negative, stage III lymph node positive, and stage IV patients, respectively (78.1% had ccRCC) [48].

## 3. Prognostic Models for Localized RCC

There are several postoperative prognostic models to assess the risk of RCC recurrence or death after surgery of localized RCC based on histopathological features, such as TNM stage, tumor size, tumor grade, coagulative necrosis, and microvascular invasion, and clinical manifestations, such as symptoms of the disease. Kattan et al. introduced the first nomogram in 2001 to assess the risk of disease recurrence for localized RCC [49], followed by the UISS, the SSIGN, the Cindolo, the Leibovich, the Sorbellini, and the Karakiewicz algorithms [50,51,52,53,54,55]. There are differences in the required prediction features and the prediction outcomes between these models. The majority (88–100%) of the patients included in these models have had clear cell RCC, although the Kattan, the UISS, the Cindolo, and the Karakiewicz models also included patients with papillary and chromophobe RCC. Because of marked differences in the histopathology and prognosis of clear cell, papillary, and chromophobe RCC subtypes, similar prediction models may not be optimal for different histological subtypes. Grading of chromophobe carcinoma is not recommended [19,27], which limits eligible prediction models for this subtype. Leibovich et al. introduced different algorithms for each histological subtype, aiming to improve the prediction accuracy. The study cohort included 75% clear cell, 17% papillary, and 6% chromophobe RCC patients [56]. Recently, Mattila et al. developed a prediction model for localized ccRCC that comprised only three features and included an easy-to-use nomogram for clinicians [57]. The properties of different prognostic models for localized RCC are described in Table 1.

The prediction accuracy (concordance index, C-index) of these prognostic models had exceeded 0.8: SSIGN 0.82–0.84 [51], Leibovich 2003 0.82, Sorbellini 0.82, Leibovich 2018 0.83–0.86, Mattila 0.76–0.84. However, these prediction models are based on the analysis of retrospective patient cohorts. A prospective validation of prediction models in a cohort of 1647 nonmetastatic (≥T1b grade 3–4 or T_any_N1M0) ccRCC patients enrolled in a sorafenib adjuvant therapy trial (ASSURE) resulted in considerably lower C-indices (0.57–0.69) for the UISS, SSIGN, Leibovich 2003, Kattan, MSKCC, Yayciogly, Karakiewicz, Cindolo, and 2002 TNM staging systems. All models demonstrated the best prediction accuracy during the first two years of follow-up after surgery [58]. Higher prediction accuracy for the first two years of follow-up was also found when comparing the Mattila and the Leibovich 2003 models: C-indices were 0.81–0.88 (Mattila) and 0.76–0.88 (Leibovich 2003) during 0–24 months and 0.78–0.84 (Mattila) and 0.71–0.82 (Leibovich 2003) during 24–90 months [57]. Late disease recurrence after 5 years of follow-up has been observed in 5–11% of patients with localized RCC [59], and the prediction of these late events remains imprecise with present prognostic models [58].

## 4. Current Applications of Biomarkers in Localized RCC

While current clinical prognostic models do not use any genetic or other biomarkers, several genetic alterations have been described for RCC. In ccRCC, the inactivation of the von Hippel–Lindau (VHL) gene is the best-described and most widely occurring genetic change seen in most sporadic ccRCCs. The inactivation of the VHL tumor suppressor can occur by numerous point mutations (over 150 described) or by suppressing transcription by methylation of the promoter areas. The inactivation of VHL function results in the activation of hypoxia-inducible transcription factors (HIF-1a and -2a) of the cellular oxygen sensing pathway, leading to the up- or downregulation of over 300 genes. These include the upregulation of proangiogenic genes, such as vascular endothelial growth factor (VEGF) [60]. In particular, HIF-2a has been shown to drive a more aggressive phenotype in ccRCC [61,62]. Since VHL inactivation has been detected from 80% to nearly all ccRCCs and is the first and universal genetic alteration in ccRCC [63,64], it does not function as a prognostic factor.

Further analyses of tumor mutations and gene expression profiles have revealed genetic features associated with prognosis in localized ccRCC. In addition to loss of VHL function, mutations in tumor suppressor genes PBRM1, BAP1, and SETD2, which function as chromatin and histone modifiers, and the PI3K/AKT pathway have been identified in nephrectomy specimens included in the Cancer Genome Atlas [65,66]. PBRM1 and BAP1 mutations have been associated with unfavorable prognosis in ccRCC [67,68]. Patients with PBMR1 or BAP1 loss had increased risk of death from RCC but it was not statistically significant after adjusting for the SSIGN score [67]. The association of gene expression profiles and RCC survival has been studied widely. A scoring system based on 16 genes discovered in gene expression analysis was observed to predict disease recurrence in localized clear cell RCCs that were stratified by stage and adjusted for tumor size, tumor grade, and the Leibovich score [69], and its prognostic ability has been validated among stage III ccRCC patients in the sunitinib adjuvant therapy trial [70]. Another gene expression signature biomarker (ClearCode34) was developed to classify good- and poor-risk clear cell RCCs and was significantly associated with RFS, OS, and CSS [71]. The cell cycle proliferation (CCP) score assay, which measures the activation of 31 genes involved in cellular proliferation, was observed to be an independent predictor of disease recurrence after nephrectomy in 565 localized RCC patients (81% ccRCC) and it outperformed the prediction accuracy of the Karakiewicz nomogram (C-index 0.87 vs. 0.84) [72].

Cell-free circulating tumor DNA (ctDNA) is a potential prognostic biomarker in multiple cancer types. Fragments of tumor DNA are released into circulation after tumor cell death and by active secretion. ctDNA can be detected from body fluids (plasma, pleural effusion, ascites, cerebrospinal fluid, and urine) with multiple methods, including polymerase chain reaction-based assays, such as droplet digital PCR (ddPCR), or next-generation DNA sequencing (NGS) [73]. Plasma or urine samples containing ctDNA fragments are easy to collect and liquid biopsy is particularly valuable when invasive tumor biopsy is not feasible or there is only a limited amount of tumor tissue available. Moreover, ctDNA may reflect heterogeneous tumor mutations better than single-site tumor biopsy and reveal therapeutically actionable mutations. Elevated ctDNA levels may reveal disease recurrence/progression before radiologically detected disease progression and thus molecular residual disease is a compelling biomarker to monitor disease recurrence after radical surgery for the primary tumor. Detectable ctDNA (molecular residual disease) has been shown to predict disease recurrence after radical surgery for localized cancer in multiple tumor types, including melanoma [74,75], colorectal cancer [76,77], and NSCLC [78,79]. Interestingly, detectable plasma ctDNA was found to be a predictive biomarker for adjuvant atezolizumab therapy after surgery of urothelial carcinoma [80]. However, a sufficient amount of ctDNA has to be present to cross the detection limit.

CtDNA has also been analyzed from plasma and urine samples of RCC patients, although studies are still scarce compared to NSCLC, colorectal cancer, melanoma, and urothelial cancer, and have mostly been done in metastatic RCC patients. Patients with metastatic ccRCC had higher plasma levels of cell-free DNA compared to localized ccRCC and healthy control patients, and higher plasma cell-free DNA levels predicted disease recurrence after nephrectomy [81]. Untargeted sequencing methods revealed detectable ctDNA in plasma or urine samples of 30–40% of RCC patients with localized and metastatic disease, and detectable ctDNA in plasma, but not in urine, was more common in patients with larger tumors and with venous tumor thrombus [82]. The rate of detectable ctDNA in RCC patients has varied markedly based on the method used (NGS panel) and patient cohort (localized or metastatic). Targeted analysis of ctDNA using an RCC-targeted NGS panel (including BAP1, KDM5C, MET, MTOR, PBRM1, PIK3CA, PTEN, SETD2, TP53, and VHL genes) revealed detectable plasma ctDNA in only 18.6% of the patients (mostly metastatic ccRCC) [82]. CtDNA analysis of plasma samples from 220 patients with metastatic RCC with a 74-gene panel revealed genomic alterations in 79% of the patients. The most frequently observed mutations included TP53 (35%), VHL (23%), EGFR (17%), NF1 (16%), and ARID1A (12%) [83]. In a smaller series of metastatic RCC patients (76% ccRCC), 18/34 (53%) of the patients had detectable plasma ctDNA and it was associated with tumor burden (the sum of longest diameter of all measurable lesions) but not with IMDC risk groups or tumor histology [84].

Upregulated programmed death ligand-1 (PD-L1 or B7-H1) expression on the surface of tumor cells is an important mechanism of tumor immune evasion. The interaction of PD-L1 and PD-1 receptors in tumor-infiltrating lymphocytes (especially cytotoxic T cells) hampers the immune response against cancer cells [85]. Although different studies have used variable methods to define PD-L1 positivity in RCC (different antibodies in immunohistochemistry, tumor cell or immune cell positivity, positivity cut-off %), PD-L1 expression has unequivocally been an adverse prognostic feature. PD-L1 expression can be found in tumor cells and in tumor-infiltrating lymphocytes (TILs) and both features have been associated with inferior survival in RCC [86]. PD-L1-positive tumor cells have been observed in 20–24% of ccRCCs and the 5-year cancer-specific survival rate of these patients was 42–47%, compared to 66–83% in PD-L1-negative patients [87,88].

In addition to a higher stage and higher tumor grade, sRCCs are found to have increased PD-L1 expression compared to ccRCCs. Genomic amplifications at 9p24.1 are more frequently found in sRCC tumors (6%) compared to ccRCC tumors (0.6%). These amplifications included JAK2, PD-L1, and PD-L2 genes, leading to upregulated PD-L1 expression [89]. In the IMmotion151 trial evaluating bevacizumab and atezolizumab vs. sunitinb in first-line metastatic RCC patients, sarcomatoid features were found in 16% (142/915) of patients. In addition, 61% of sarcomatoid RCCs (86/142) were PD-L1-positive (≥1% tumor-infiltrating immune cells positive), compared to 40% of PD-L1-positive cases among all study patients (362/915) [90,91]. In the CheckMate 214 trial evaluating ipilimumab and nivolumab vs. sunitinib in treatment-naive metastatic ccRCC patients, 13% of all patients (145/1096) had sarcomatoid features and only 4% (6/145) had an IMDC favorable risk score. Of 139 sRCC patients with IMDC intermediate or poor risk scores, 50% were PD-L1-positive (≥1% tumor cells positive), compared to 26% of all IMDC intermediate- or poor-risk patients [92]. This feature renders sRCCs more susceptible to ICI than to antiangiogenic TKI therapies, and the introduction of ICI has significantly improved treatment outcomes in patients with advanced sRCC [90,92].

In addition to gene expression profiles, ctDNA, and PD-L1 expression levels, the prognostic ability of epigenetic biomarkers, such as DNA methylation, expression of microRNAs, and long noncoding RNA, is being studied. Cell-free DNA methylation analysis from plasma and urine samples has been introduced as a potential method detect early-stage RCC patients from among healthy control patients [93]. However, none of these biomarkers are yet recommended in the international RCC guidelines [4,94], nor have they been adopted into widespread clinical use. The aim of future studies is to supplement current prognostic algorithms with novel biomarkers to improve their prediction accuracy and validate these findings in independent patient cohorts.

## 5. Prognostic Markers and Adjuvant Therapies for Localized ccRCC

The efficacy of antiangiogenic TKI therapies and immune checkpoint inhibitors (ICI) in the treatment of advanced ccRCC has led to adjuvant therapy trials aiming to reduce the risk of disease recurrence and improve the overall survival (OS) of patients with localized RCC after radical or partial nephrectomy. Before TKI and ICI therapies, cytokines (interferon-alpha and high-dose interleukin-2) showed modest clinical activity (response rates of 15–31%) in stage IV RCC [95] and were also studied in the adjuvant setting. However, cytokine and tumor vaccine adjuvant therapy trials failed to improve recurrence-free and overall survival [94,95,96,97,98].

The next attempt to improve RFS and OS was made with VEGF-targeted TKI adjuvant therapies. Five large, prospective, multicenter trials with sunitinib (S-TRAC), sunitinib and sorafenib (ASSURE), pazopanib (PROTECT), axitinib (ATLAS), and sorafenib (SORCE) were conducted [99,100,101,102,103]. The design of adjuvant therapy trials and results are described in Table 2. There were various inclusion criteria for intermediate- and high-risk patients and the proportion of higher-risk (≥T3 or N1) patients was different across these adjuvant trials. The inclusion criteria for the S-TRAC trial were modified from the UISS criteria (T3N0M0 Fuhrman grade ≥ 2 and ECOG performance status ≥ 1, T4N0M0 any Fuhrman grade, any ECOG PS, or T_any_N1-2M0). The ASSURE and the PROTECT trials required Fuhrman grade ≥ 3 for lower-risk (T1b-T2) tumors, whereas the ATLAS trial included >T2 tumors regardless of Fuhrman grade. The SORCE trial was the only trial that directly adopted the existing prognostic algorithm (the Leibovich score (2003)) for classifying patients into intermediate- (3–5 points) or high-risk (6–11 points) groups for disease recurrence. The proportion of lower-risk patients (T1-2, stage I and II) ranged from 11% to 35% in the ATLAS, PROTECT, ASSURE, and SORCE trials [100,101,102,103].

All adjuvant TKI trials were placebo-controlled and aimed to show the DFS benefit, but only S-TRAC yielded a positive result, with a 1.2-year improvement in the DFS of the sunitinib arm. Tumor cell PD-L1 expression was not statistically significantly associated with DFS, whereas high tumor CD8+ T-cell density was predictive for longer DFS in the sunitinib arm of the S-TRAC trial [105]. The S-TRAC, PROTECT, and ATLAS trials included only ccRCC patients, and the majority (79% and 84%) of patients enrolled in the ASSURE and SORCE trials had ccRCC. Usually, the protocol-specified duration of adjuvant TKI therapy was 12 months. The ATLAS and the SORCE trials included cohorts with adjuvant TKI therapy up to 36 months, but the longer duration of TKI therapy did not lead to improved DFS. Adjuvant TKI therapy caused substantial toxicity (grade 3–4 adverse events 49–72%) and a significant proportion of the patients (23–49%) discontinued adjuvant TKI therapy because of intolerable toxicity or refused to continue study therapy (96–100). Currently, adjuvant TKI therapy is not recommended after complete resection of the primary tumor in the international RCC guidelines due to the substantial toxicity and the lack of OS benefit [4,94].

Immune checkpoint inhibitors (ICI) have replaced cytokines in the immune therapy of advanced RCC and are also being studied in randomized placebo-controlled prospective clinical trials in the adjuvant and neoadjuvant setting. IMmotion010 is evaluating 12-month adjuvant therapy with PD-L1 inhibitor atezolizumab, PROSPER neoadjuvant therapy (nivolumab two doses), followed by 9-month adjuvant therapy with PD-1 inhibitor nivolumab and CheckMate 914 6-month adjuvant therapy with the combination of CTLA-4 inhibitor ipilimumab and PD-1 inhibitor nivolumab in resected localized ccRCC patients, and RAMPART 12-month durvalumab adjuvant therapy and 12-month adjuvant CTLA-4 and PD-L1 inhibitor (tremelimumab and durvalumab) combination therapy. The first results of these trials are expected to be published in 2022–2024. The results from the KEYNOTE-564 trial evaluating 12-month adjuvant therapy with pembrolizumab in resected intermediate- or high-risk ccRCC patients showed a statistically significantly longer recurrence-free survival rate in the pembrolizumab arm compared to the placebo arm at 24 months (77.3% vs. 68.1%, HR for recurrence or death 0.68 (0.53–0.87)) (Table 2) [105]. As this was the first analysis, a longer follow-up will be needed to confirm the survival outcomes of the pembrolizumab adjuvant therapy. However, ICI may finally become a practice-changing adjuvant treatment option for RCC patients after complete resection of the primary tumor and lymph node or distant metastases.

## 6. Discussion

Numerous traditional histopathological factors and an increasing number of biomarkers have been identified to affect the postoperative prognosis of patients with localized ccRCC. The individual assessment of the risk for disease recurrence after radical or partial nephrectomy is important to tailor the intensity of postoperative follow-up imaging. Moreover, accurate risk assessment for disease recurrence is essential to select optimal patients for adjuvant therapy trials. However, there is no consensus regarding which is the best model or biomarker to choose to guide the clinical decision making. Limitations in the availability of biomarker analyses, time required to obtain the results, costs from the analyses, and, in particular, the lack of sufficient clinical validation still limit the use of prognostic biomarkers in clinical practice. Useful risk assessment tools for clinicians should be easy-to-use and include only a moderate amount of readily available risk factors (e.g., 3–5 traditional histopathological factors). Different clinicopathological features may be available in different centers. In the future, biomarkers, including those from plasma and urine (liquid biopsies), may supplement these prognostic algorithms.

Prognostic models with traditional histopathological and clinical factors should be easy-to-use and readily available. However, only the SORCE trial had incorporated a prognostic algorithm into the inclusion criteria of the trial. Adjuvant TKI trials underscored the fact that careful patient selection is required to avoid substantial toxicity and enrich higher-risk patients for adjuvant therapy. A meta-analysis of adjuvant TKI trials showed a DFS benefit in the high-risk (T3, Fuhrman grade 3–4; T4, or N1) population but not in the low-risk population (pooled HR for DFS 0.85 (0.75–0.97) and 0.98 (0.82–1.17), respectively) [106]. The optimal selection criteria for the high-risk localized ccRCC population remain to be defined. The results from the biomarker analyses of current neoadjuvant and adjuvant trials with ICI may shed more light on the issue.

## 7. Conclusions

Prognostic factors and validated prediction models help to evaluate the risk for disease recurrence after complete surgical resection of localized ccRCC. Better models to reduce follow-up imaging in low-risk patients and optimize the selection of patients for adjuvant trials are required. The combination of clinical and histopathological features with novel biomarkers may improve the prediction accuracy of prognostic models. The optimal set of prognostic factors and biomarkers to define high-risk patients for disease recurrence may be discovered in ongoing placebo-controlled randomized prospective clinical trials.

## Figures and Tables

**Table 1 cancers-14-00239-t001:** Postoperative prognostic models for localized RCC.

Reference	RCC Subtype	Prediction Outcome	Number of Risk Groups	Prediction Features	Number of Patients
Kattan (2001) [49]	Clear Cell, Papillary, and Chromophobe RCC	Recurrence-Free Survival	Not Defined	Symptoms (Incidental, Local, Systemic), Histology, Tumor Size, 1997 T Stage	612
UISS (2001) [50]	Clear Cell, Papillary, and Chromophobe RCC	Overall Survival	5	1997 TNM Stage, Fuhrman Grade, ECOG Performance Status	661
SSIGN (2002) [51]	Clear Cell RCC	Cancer-Specific Survival	10	1997 TNM Stage, Tumor Size (<5 cm, ≥5 cm), Tumor Grade, Necrosis	1801
Cindolo (2003) [52]	Clear Cell, Papillary, and Chromophobe RCC	Recurrence-Free Survival	Not Defined	Symptoms (Asymptomatic, Symptomatic), Tumor Size	660
Leibovich (2003) [53]	Clear Cell RCC	Metastasis-Free Survival	8 (0–2 low, 3–5 Intermediate, ≥6 High)	2002 TNM Stage, Regional Lymph Node Involvement	479
Sorbellini MSKCC (2005) [54]	Clear Cell RCC	Recurrence-Free Survival	Not Defined	Tumor Size, 2002 TNM Stage, Fuhrman Grade, Necrosis, Microvascular Invasion, Presentation (Incidental, Local Symptoms, Systemic Symptoms)	701 + Validation Cohort 200
Karakiewicz (2007) [55]	Clear Cell, Papillary, and Chromophobe RCC	Cancer-Specific Survival	Not Defined	2002 TNM Stage, Tumor Size, Fuhrman Grade, Symptoms (Non, Local, Systemic)	2530 + Validation Cohort 1377
Leibovich (2018) [56]	Clear Cell, Papillary, and Chromophobe RCC	Progression-Free and Cancer-Specific Survival	19	Constitutional Symptoms (Yes, No), Tumor Grade, Coagulative Necrosis, Sarcomatoid Differentiation, Tumor Size, Perinephric or Renal Sinus Fat Invasion, Tumor Thrombus Level, Extension Beyond Kidney, and Nodal Involvement	3633
Mattila (2021) [57]	Clear Cell RCC	Metastasis-Free Survival	3 (Low, Intermediate, High)	Tumor Size, Fuhrman Grade, Microvascular Invasion	196 + Validation Cohort 714

**Table 2 cancers-14-00239-t002:** The results from phase III randomized adjuvant TKI and ICI trials in RCC.

Trial	Treatment	Inclusion Criteria	Median DFS/HR of Disease Recurrence or Death	Discontinuation Rate Due to AE/(AE + Patient Withdrawal) ^#^
S-TRAC [99]	Sunitinib vs. Placebo 12 Months	≥T3N0 (gr ≥ 2, ECOG ≥ 1) or T_any_N1	6.8 Years, HR 0.76 (0.59–0.98) vs. 5.6 Years	28% vs. 6%
ASSURE [100]	Sunitinib vs. Sorafenib vs. Placebo 12 Months	≥T1b (gr 3–4) N0 or T_any_N1	5.8 Years, HR 1.17 (0.90–1.52) vs. 6.1 Years, HR 0.97 (0.75–1.28) vs. 6.6. Years	44% ^#^ vs. 45% ^#^ vs. 11% ^#^
PROTECT [101]	Pazopanib vs. Placebo 12 Months	T2 (gr 3–4) N0, T3–4N0, or T_any_N1	HR 0.86 (0.70–1.06)	35% vs. 5%
ATLAS [102]	Axitinib vs. Placebo 12–36 Months	≥T2N0 or T_any_N1	HR 0.87 (0.660–1.147)	23% vs. 11%
SORCE [103]	Sorafenib 12 Months vs. Sorafenib 36 Months vs. Placebo	Intermediate Risk (Score 3–5) or High Risk (Score ≥ 6) According to Leibovich (2003)	HR 0.94 (0.77–1.14) Sorafenib 12 Months vs. Placebo HR 1.01 (0.82–1.23) Sorafenib 36 Months vs. Placebo	44% ^#^ vs. 49% ^#^ vs. 12%
KEYNOTE-564 [104]	Pembrolizumab vs. Placebo 12 Months	T2 (gr 3–4 or Sarcomatoid) N0, T3–4N0, T_any_N1, or Resected M1	HR 0.68 (0.53–0.87)	21% vs. 2%

^#^ indicates AE + patient withdrawal.

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
