# Peer review of "Prognostic Factors for Localized Clear Cell Renal Cell Carcinoma and Their Application in Adjuvant Therapy"

_cancers, 2022, doi:10.3390/cancers14010239_

Round 1
Reviewer 1 Report
The authors should be commended for a thorough review on localized clear cell renal cell carcinoma and their application in adjuvant therapy.
The review is detailed and covers all the important points.
I do think the authors should mention the current ongoing adjuvant/perioperative immune checkpoint trials that will probably change the landscape of treatment in this disease space in the near future:
- IMmotion 010 (Atezolizumab as adjuvant therapy -NCT03024996)
- cHECKMATE 914 - Nivolumab vs. Nivo+Ipi and placebo - NCT03138512)
- Prosper (Nivo in patients with localized kidney cancer undergoing nephrectomy - NCT03055013)
- RAMPART - NCT03288532
I think it is also IMPORTANT to describe the growing role of immunotherapy in RCC with sarcomatoid tumors.
Author Response
The RAMPART trial was added to ongoing neoadjuvant/adjuvant RCTs in the section 4 (Prognostic markers and adjuvant therapies for localized ccRCC), lines 378−380.
“IMmotion010 is evaluating 12-month adjuvant therapy with PD-L1 inhibitor atezolizumab, PROSPER neoadjuvant therapy (nivolumab two doses) followed by 9-month adjuvant therapy with PD-1 inhibitor nivolumab and CheckMate 914 6-month adjuvant therapy with the combination of CTLA-4 inhibitor ipilimumab and PD-1 inhibitor nivolumab in resected localized ccRCC patients, and RAMPART 12-month durvalumab adjuvant therapy and 12-month adjuvant CTLA-4 and PD-L1 inhibitor (tremelimumab and durvalumab) combination therapy.”
The role of immunotherapy in sarcomatoid RCC was discussed in the section 3 (Current applications of biomarkers in localized RCC), lines 321−322.
“Of 139 sRCC patients with IMDC intermediate or poor risk score, 50% were PD-L1 positive (≥1% tumor cells positive) compared to 26% of all IMDC intermediate or poor risk patients (90). This feature renders sRCCs more susceptible to ICI than to antiangiogenic TKI therapies and the introduction of ICI have significantly improved treatment outcomes in patients with advanced sRCC [91,93].”
Reviewer 2 Report
well written review about prognostic factors in localized RCC.
Major: missing important info in this review ie:
https://pubmed.ncbi.nlm.nih.gov/29773662/
https://pubmed.ncbi.nlm.nih.gov/29374054/
https://pubmed.ncbi.nlm.nih.gov/32572266/
Minor comments.
sarcomatoid features may be seen in any type or RCC
reference 11 is overcited. I would try to add info from other sources rather citing 11
I would suggest to make subjective recommendations. Which prognostic model the authors would recommend? With biomarker consider the most interest?
Author Response
These studies were added to manuscript to the section 3 (Current applications of biomarkers in localized RCC) and 4 (Prognostic markers and adjuvant therapies for localized ccRCC).
Lines 250−251.
“A scoring system based on 16 genes discovered in gene expression analysis was observed to predict disease recurrence in localized clear cell RCCs that were stratified by stage and adjusted for tumor size, tumor grade, and the Leibovich score (68), and it`s prognostic ability has been validated among stage III ccRCC patients in the sunitinib adjuvant therapy trial [71].”
Lines 358−360.
“All adjuvant TKI trials were placebo controlled and aimed to show the DFS benefit but only S-TRAC yielded positive result with 1.2-year improvement in the DFS of sunitinib arm. Tumor cell PD-L1 expression was not statistically significantly associated with DFS, whereas high tumor CD8+ T-cell density was predictive for longer DFS in the sunitinib arm of S-TRAC trial [105].”
Lines 325−327.
“In addition to gene expression profiles, ctDNA, and PD-L1 expression levels, the prognostic ability of epigenetic biomarkers, such as DNA methylation, expression of microRNAs, and long noncoding RNA are being studied. Cell free DNA methylation analysis from plasma and urine samples has been introduced as a potential method detect early stage RCC patients from healthy control patients [94].”
We have discussed sarcomatoid features in manuscript, lines 100−102.
"Approximately 5% of RCCs undergo epithelial to mesenchymal transition and present with sarcomatoid differentiation and sarcomatoid features have been observed in clear cell, papillary, and chromophobe RCCs [12]."
References were updated and corrected as suggested, lines 100−132:
- Delahunt, B.; Srigley, J.R.; Egevad, L.; Montironi, R.; International Society for Urological Pathology. International Society of Urological Pathology Grading and Other Prognostic Factors for Renal Neoplasia. Eur. Urol. 2014, 66, 795-798.
- Blum, K.A.; Gupta, S.; Tickoo, S.K.; Chan, T.A.; Russo, P.; Motzer, R.J.; Karam, J.A.; Hakimi, A.A. Sarcomatoid Renal Cell Carcinoma: Biology, Natural History and Management. Nat. Rev. Urol. 2020, 17, 659-678.
- Cheville, J.C.; Lohse, C.M.; Zincke, H.; Weaver, A.L.; Leibovich, B.C.; Frank, I.; Blute, M.L. Sarcomatoid Renal Cell Carcinoma: An Examination of Underlying Histologic Subtype and an Analysis of Associations with Patient Outcome. Am. J. Surg. Pathol. 2004, 28, 435-441.
- Delahunt, B.; Cheville, J.C.; Martignoni, G.; Humphrey, P.A.; Magi-Galluzzi, C.; McKenney, J.; Egevad, L.; Algaba, F.; Moch, H.; Grignon, D.J. et al. The International Society of Urological Pathology (ISUP) Grading System for Renal Cell Carcinoma and Other Prognostic Parameters. Am. J. Surg. Pathol. 2013, 37, 1490-1504.
- Delahunt, B.; McKenney, J.K.; Lohse, C.M.; Leibovich, B.C.; Thompson, R.H.; Boorjian, S.A.; Cheville, J.C. A Novel Grading System for Clear Cell Renal Cell Carcinoma Incorporating Tumor Necrosis. Am. J. Surg. Pathol. 2013, 37, 311-322.
- Moch H, Humphrey PA, Ulbright TM, Reuter VE. WHO Classification of Tumours of the Urinary System and Male Genital Organs. WHO Classification of Tumours, 4th Edition, Volume 8 (2016).
We have extended our proposed subjective recommendation, lines 400−404:
“However, there is no consensus, which is the right model or biomarker to choose to guide the clinical decision making. Limitations in the availability of biomarker analyses, time required to get the results, costs from the analyses, and in particular the lack of sufficient clinical validation still limit the use of prognostic biomarkers in clinical practice. Useful risk-assessment tools for clinicians should be easy-to-use and include only moderate amount of readily available risk factors (e.g. 3−5 traditional histopathological factors). Different clinicopathological features may be available in different centers. In the future, biomarkers including those from plasma and urine (liquid biopsies) may supplement these prognostic algorithms.”
Reviewer 3 Report
The topic of adjunct treatment for loclaized renal cell carcinoma is well presented with emphasis on tumor marlers and clinical trials ongoing.
Could you present specific guideines upon to treat a oatient after nephrectom and if so: which would be the therapy (torosine kinases, imune checkpoin inhibitors or both).
Author Response
In our opinion, it´s too early to make guidelines for choosing adjuvant therapy for RCC patients after surgery of the primary tumor. Adjuvant TKI therapy is not recommended by the international guidelines (NCCN, ESMO). Longer follow-up is needed to confirm the benefit of adjuvant ICI in RCC and the results from RCTs evaluating neoadjuvant/adjuvant combination therapies will be presented in the future. This issue is discussed in the section 4 (Prognostic markers and adjuvant therapies for localized ccRCC), lines 355−387.
“Currently, adjuvant TKI therapy is not recommended after complete resection of the primary tumor in the international RCC guidelines due to substantial toxicity and the lack of OS benefit [4,95]. “
“Immune checkpoint inhibitors (ICI) have replaced cytokines in the immune therapy of advanced RCC and are also being studied in randomized placebo controlled prospective clinical trials in the adjuvant and neoadjuvant setting … First results of these trials are expected to be published in 2022−2024. The results from KEYNOTE-564 trial evaluating 12-month adjuvant therapy with pembrolizumab in resected intermediate- or high-risk ccRCC patients showed statistically significantly longer recurrence-free survival rate in pembrolizumab arm compared to placebo arm at 24 months (77.3% vs 68.1%, HR for recurrence or death 0.68 (0.53−0.87) (Table 2) (101). As this was the first analysis, longer follow-up will be needed to confirm the survival outcomes of the pembrolizumab adjuvant therapy. However, ICI may finally become practice-changing adjuvant treatment option for RCC patients after complete resection of the primary tumor and lymph node or distant metastases.”
We have also extended our proposed subjective recommendation, lines 400−404:
“However, there is no consensus, which is the right model or biomarker to choose to guide the clinical decision making. Limitations in the availability of biomarker analyses, time required to get the results, costs from the analyses, and in particular the lack of sufficient clinical validation still limit the use of prognostic biomarkers in clinical practice. Useful risk-assessment tools for clinicians should be easy-to-use and include only moderate amount of readily available risk factors (e.g. 3−5 traditional histopathological factors). Different clinicopathological features may be available in different centers. In the future, biomarkers including those from plasma and urine (liquid biopsies) may supplement these prognostic algorithms.”